# The Flipping-Wedge Osteotomy: How 3D Virtual Surgical Planning (VSP) Suggested a Simple and Promising Type of Osteotomy in Pediatric Post-Traumatic Forearm Deformity

**DOI:** 10.3390/jpm13030549

**Published:** 2023-03-19

**Authors:** Alessandro Depaoli, Grazia Chiara Menozzi, Giovanni Luigi Di Gennaro, Marco Ramella, Giulia Alessandri, Leonardo Frizziero, Alfredo Liverani, Daniela Martinelli, Gino Rocca, Giovanni Trisolino

**Affiliations:** 1Unit of Pediatric Orthopedics and Traumatology, IRCCS Istituto Ortopedico Rizzoli, 40136 Bologna, Italy; 2Department of Industrial Engineering, Alma Mater Studiorum University of Bologna, 40136 Bologna, Italy; 3Unit of Pediatric Orthopedics and Traumatology, Nuovo Ospedale “Santo Stefano”, 59100 Prato, Italy

**Keywords:** forearm deformity, post-traumatic, malunion, osteotomy, VSP, computer-aided surgery, pediatric

## Abstract

(1) Background: The application of computer-aided planning in the surgical treatment of post-traumatic forearm deformities has been increasingly widening the range of techniques over the last two decades. We present the “flipping-wedge osteotomy”, a promising geometrical approach to correct uniapical deformities defined during our experience with virtual surgical planning (VSP); (2) Methods: a case of post-traumatic distal radius deformity (magnitude 43°) treated with a flipping-wedge osteotomy in an 11-year-old girl is reported, presenting the planning rationale, its geometrical demonstration, and the outcome of the procedure; (3) Results: surgery achieved correction of both the angular and rotational deformities with a neutral ulnar variance; (4) Conclusions: flipping-wedge osteotomy may be a viable option to achieve correction in forearm deformities, and it deserves further clinical investigation.

## 1. Introduction

Currently, acute correction of post-traumatic forearm deformities in children is generally achieved with a closing-wedge osteotomy, an opening-wedge osteotomy, or an oblique osteotomy of one or both bones [1,2]. The precision of these techniques has significantly improved since the introduction of virtual surgical planning (VSP) and the spread of 3D-printed patient-specific instruments (PSIs) in orthopedic surgery [1,2,3,4]. Many authors have published methods and algorithms to precisely align a malunited bone to the shape of the uninjured contralateral with a single oblique cut [5,6,7,8,9]. However, this method still has some issues: (1) VSP is still technically demanding; (2) it heavily relies on PSI-guided cuts that are difficult to achieve freehand, even by the most expert surgeons; (3) rotational correction must be preoperatively calculated to set the cutting plane; and (4) if the rotational deformity to correct approaches zero, a very steep oblique cut is needed, which may not be clinically feasible [3].

In recent years, we have treated several congenital and post-traumatic deformities using VSP and 3D-printed PSIs [10,11,12,13]. In particular, we reported a case of forearm malunion in a 15-year-old boy treated by repositioning a local bone wedge [11]. In this case, the correction was obtained with an empirical approach, and the wedge was rotated upside down. By approaching a similar case, we developed a reliable method to shape a trapezoidal wedge that can straighten the bone by flipping it 180° on its longitudinal axis. In this article, we present a case of post-traumatic distal radius deformity in an 11-year-old girl treated with a “flipping-wedge osteotomy” and its geometrical demonstration. 

## 2. Materials and Methods

### 2.1. Case Presentation

An 11-year-old girl came to our attention for a post-traumatic deformity of the distal radius on the dominant side. Approximately eight months earlier, she had suffered a fracture of the distal meta-diaphysis of the radius and distal ulna that was treated elsewhere with open reduction, percutaneous pinning, and casting. After pin removal, the distal radius developed malunion, with 35° of flexion and 25° of radial angulation (43° of magnitude on the true plane). Upon physical examination, the patient showed an evident deformity, with flexion, angulation, and pronation of the distal forearm (Figure 1). The overall arc of rotation was 60° (10° in further pronation and 50° in supination). Flexion and extension of the wrist were slightly reduced compared to the contralateral wrist (60° and 70°, respectively). No neurovascular impairment was observed in her hand and fingers.

### 2.2. Surgical Simulation and Planning

Through a well-established process named segmentation, a 3D digital bone model was obtained from the CT scans of both forearms [10]. Some surgical plannings were simulated to empirically align the malunited radius to the uninjured side by reallocating a local bone wedge, as already described [11]. During these attempts, we observed that the angle of the deformity could be corrected by rotating a bone wedge 180° on its longitudinal axis with a trapezoid section constructed as follows: the major leg was obtained by cutting the bone along the transverse bisector line (tBL) at the center of rotation of angulation (CORA) between axes, and the minor leg was obtained by cutting the bone perpendicularly to one of the axes at any desired distance (see Figure 2 and Appendix A) [14]. The crucial element for this approach is the accurate assessment of the deformity, particularly the identification of the plane of maximal deformity (also called the true plane or plane of maximal angulation) [14]. In fact, as described by Paley, angular deformities present on anteroposterior (AP) and laterolateral (LL) radiographs are actually uniplanar in an oblique plane [14]. In the 3D model, we were also able to choose the best plate and screw configuration that could spare the distal radial growth plate.

### 2.3. Geometric Demonstration

**Theorem** **1.** 
*Any unifocal bone deformity can be corrected by rotating a trapezoidal bone wedge 180° with one cut along the transverse bisector line (tBL) at the center of rotation of angulation (CORA) and one cut perpendicular to any one of the axes at any distance from the CORA. Any further rotational malalignment can be corrected at the latter site of the osteotomy.*


**Proof** **of** **Theorem** **1.** Between two lines *a* and *b* incident at the CORA, which represent bone axes (Figure 3a), two couples of opposite angles are identified on the plane both axes lay on: one couple is defined as the angular deformity (α), and the other as its supplementary angle (β = 180° − α, Figure 3b). During this phase, rotational deformity between axes is not considered since it will be addressed later. The point of intersection between axes represents the CORA. A “concavity angle” (γ) and a “convexity angle” (δ) are identified and calculated: 

γ = β = 180° − α 

δ = 2 α + β = 2 α + (180° − α) = 180° + α

To correct the deformity, both γ and δ need to equal 180°. To achieve this, we need to bring the side of the *b* axis below the CORA parallel to the *a* axis. This can be achieved by mirroring or flipping by 180° the *b* axis along the longitudinal bisector line (lBL). We must consider only the lower part to the CORA in order to apply this reasoning to a bone that has to be cut. This is why we considered the transverse bisector line (tBL), which is obviously perpendicular to the lBL, as the “cutting plane” in order to flip only the plane side below it. Once flipped, we obtain γ’ and δ’ both equal to 180°, and axes *a* and *b* overlap.
γ’ = γ + α = 180° − α + α = 180°
δ’ = δ − α = 180° + α − α = 180°

These considerations are defined in two dimensions, but they can be contextualized in a three-dimensional system to plan the bone correction by applying them on the plane of maximal deformity in which the deformity reaches the highest magnitude [14]. The axes and the CORA were identified according to the previous reasoning (Figure 3a,b). Idealizing the bone as a rectangle with sides parallel to the axes, we can again apply the corresponding angles theorem, obtaining the corresponding angles, as shown in Figure 3c. Cutting along the tBL will correct the deformity, while the second cut of the bone wedge needs to be perpendicular to one of the axes, at any distance preferred by the surgeon, because it does not affect the correction (Figure 3d,e).

Finally, if rotational deformities are also present, they can be corrected by rotating the diaphysis that is on the side where the perpendicular cut was made (Figure 3f). Obviously, it is possible to choose where to do the perpendicular cut, proximal or distal to the cut on the bisector. □

### 2.4. Surgical Treatment

The patient underwent surgical treatment with a volar Henry’s approach to the concave side of the deformity. Patient-specific cut guides were not available for technical issues; however, due to the important angle of deformity, the tBL was easy to identify intraoperatively with fluoroscopy and a K-wire (Figure 4). After positioning another K-wire perpendicular to the proximal bone segment, cuts were made with an oscillating saw according to the preoperative planning, with approximately 1 cm of distance between them (Figure 4). Before separating the bone fragments, a longitudinal mark was drawn to monitor rotational alignment. The trapezoidal bone wedge was then flipped 180° as planned and then fixed with a 3.5 mm PediFrag locking T-plate with 2 + 5 holes (OrthoPaediatrics, Warsaw, IN, USA). The residual rotational deformity of approximately 40° between the wedge and the proximal segment was intraoperatively corrected (Figure 4). A brachio-metacarpal cast was then applied for 30 days, and drainage was maintained for 24 h postoperatively. There were no postoperative complications. After cast removal, the patient underwent a rehabilitation protocol, and a wrist splint was prescribed for 40 more days.

## 3. Results

At the 5-month follow-up, the patient showed a range of motion of the right wrist of 70° of extension and 80° of flexion and prono-supination of the right forearm of 60–0–80° compared to 80–0–90° of the uninjured side (Figure 5). Radiographs showed complete healing of the osteotomy with maintained alignment and neutral ulnar variance (Figure 6b).

## 4. Discussion

The conventional approach for acute correction of post-traumatic malunion deformities in children is represented by a corrective osteotomy at the deformity site (mainly closing-wedge osteotomy, opening-wedge osteotomy, or oblique osteotomy) planned on plain radiographs, with or without an additional osteotomy of the other bone, to achieve acceptable ulnar variance and satisfactory prono-supination [1,2,15,16,17,18,19,20]. Since the spread of computer-aided planning and the introduction of PSIs in the last two decades, many different options have become viable to improve surgical accuracy and final outcome [1,4,21,22,23,24,25]. Technology has made it easier to achieve correction with a single oblique osteotomy, which allows correction of both the angular and rotational deformities, preserving bone length [4,26]. 

The correction of complex bone deformities with a single oblique osteotomy was first described by Merle d’Aubigné and Vaillet in 1961 to achieve correction of both valgus and rotational deformity in the proximal femur [27]. This approach was then re-presented by many authors and applied in several districts and conditions, such as Rab osteotomy in Blount’s disease [28,29,30]. The solid mathematical bases of this method were deeply explored by Sangeorzan in 1989 and then simplified by Waanders and Herzenberg in 1992 [5,6,7]. Despite VSP and 3D-printed PSIs having improved this approach in forearm surgery, the plane of an oblique single-cut rotational osteotomy (OSCRO) calculated by algorithms may be too steep to be clinically achievable [31]. Dobbe et al. in 2021 proposed an oblique double-cut rotational osteotomy (ODCRO) calculated with software to obtain the smoothest alignment of the corrected bone when a single oblique cut is too steep [3]. The flipping-wedge osteotomy is a lengthening-preserving approach that may avoid a shortening osteotomy if one of the bones is straight. Despite the disadvantages of a double cut, in all corrections we simulated, even with severe deformities, the profile of the bone never appeared too askew.

Some authors have already described techniques for correcting upper limb deformities by rotating a local bone graft. The reverse-wedge osteotomy (RWO) presented by Dagrégorio and Saint-Cast in 2005 proposed the rotation of a metaphyseal bone wedge to treat Madelung’s deformity [32]. Mallard et al. in 2012 analyzed the application of this technique in 10 wrists in 5 patients: all osteotomies healed within 3 months, but 3 wrists required ulnar shortening osteotomies for persistent impingement [33]. Although each article highlighted the importance of preoperative planning, both gave only generic rules to perform the cut. In 2007, Yong et al. published an article in which six metacarpal malunions in five patients were successfully treated with the trapezoid rotational bone graft osteotomy [34]. This technique perfectly resembles the flipping-wedge osteotomy, but both cuts are made at one-quarter of the angle of deformity to each perpendicular to the bone, thus not allowing rotational correction and resulting in a more twisted-shaped bone. 

Several methods of deformity correction by repositioning a local wedge to avoid shortening of the bone have been reported in the literature. The inverted V-shaped osteotomy described by Levy et al. in 1973 resembled a neutral-wedge osteotomy in which genu varum is corrected by medially transferring a slice of a chevron cut [35]. In 1998, Nagi et al. presented a similar approach but with a straight transverse cut [36]. To compensate for tibial defects in total knee arthroplasty, Franceschina and Swienckowski in 1999 proposed a reversed tibial flip autograft, where an entire wedge of the proximal tibia is rotated upside down [37]. In 2009, Russell et al. described the clamshell osteotomy, a simple but effective strategy to correct even complex diaphyseal deformities by creating a multifragmentary fracture to then allow stabilization with a straight intramedullary nail [38]. 

In summary, the advantages of the flipping-wedge osteotomy are:*Preservation of bone length*: The final length of the bone is not changed by this technique. However, if shortening of the wedge is needed, a cylindrical slice of the requested length can be removed from one of the extremities.*Rotational correction can be added*: Rotational adjustment can also be combined with angular correction, since one of the osteotomies is perpendicular to the final longitudinal axis of the segment.*Clever planning method in all situations*: The flipping-wedge osteotomy allows surgeons to plan and obtain corrections, even without more advanced, personalized, and precise technologies and instruments, such as PSIs or modeling software. With a single-cut osteotomy, such as the oblique method of Sangeorzan, the risk of creating secondary deformities is very high if performed free-hand [5,6]. Although the use of PSIs can facilitate and improve the precision of the flipping-wedge osteotomy, this technique may be applied even in settings with basic equipment, as long as an accurate preoperative radiographic assessment with the exact identification of the plane of maximal deformity is performed.*Suitability for diaphyseal segments and intramedullary fixation*: This method may be a useful way to provide a straight path for an intramedullary nail. Compression of the bone segments on the graft and the use of Poller screws, when needed, allow surgeons to achieve great stability, and the graft will automatically find its most proper alignment.

However, we point out some disadvantages of this technique: *Higher risk of non-union:* Compared to a single-cut osteotomy, the fixation of an intercalary wedge increases the risk of non-union, especially if applied in adults. Yong et al., among 12 osteotomy sites in 6 cases of diaphyseal metacarpal malunion in patients aged between 20 and 51 years old treated with rotation of a local trapezoid graft, reported 2 sites of delayed union [34]. These results warn that soft tissue conditions and healing potential must be carefully considered before performing a flipping-wedge osteotomy.*Absence of a hinge:* We believe that closing- and opening-wedge osteotomies are still the most reliable and easy way to fix osteotomies until the surgeon can keep a hinge of cortical bone between them. In many districts, hinge loss usually decreases the power of correction and makes the osteotomy more difficult to fix properly without further sliding and rotational malalignments. We highlight that proper fixation of the flipping-wedge osteotomy can be challenging. However, we believe that the use of intrinsic and extrinsic strategies to increase stability (e.g., wedge cut, temporary pinning, and/or temporary external fixation) may be beneficial.*Risk of further rotational malalignment:* Rotation is the double-edged sword of this technique. Similar to a closing-wedge osteotomy without a hinge, surgeons must check rotational alignment before final fixation of the osteotomy.

## 5. Conclusions

The flipping-wedge osteotomy is a simple and geometrically demonstrated technique that allows restoration of correct bone alignment and preservation of bone length. An accurate three-dimensional assessment is crucial to identify the plane of maximal deformity and to plan the correction. We demonstrated the application of this method in a child undergoing osteotomy of the distal radius for post-traumatic malunion. However, additional cases are required to confirm the reproducibility and versatility of this method.

## Figures and Tables

**Figure 1 jpm-13-00549-f001:**
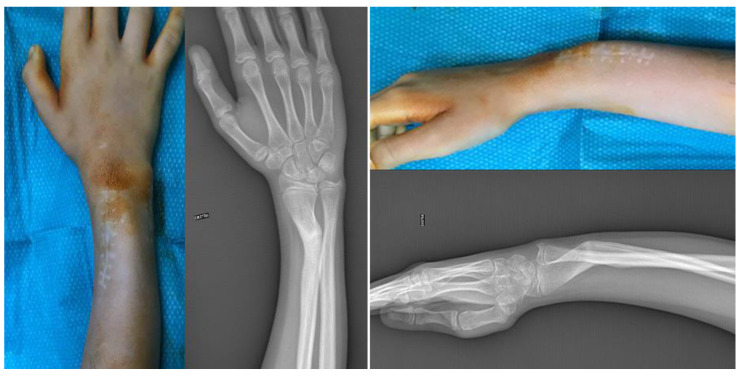
Clinical and radiographic appearance of the distal forearm. A severe angulation of the distal meta-diaphysis of the radius is visible in both anteroposterior and laterolateral projections.

**Figure 2 jpm-13-00549-f002:**
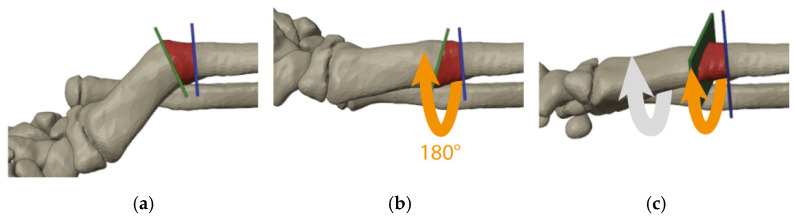
Simulation of surgical steps of the flipping-wedge osteotomy on the 3D model obtained from the CT scan with the radius at the plane of maximal angulation: (**a**) one cut is made along the transverse bisector line (green plane) and the other perpendicular to the longitudinal axis of the proximal bone (blue plane); (**b**) the bone wedge within cuts (red) is then flipped 180° on the longitudinal axis (orange arrow); and (**c**) further correction of the rotational deformity is achieved by rotating all segments distal to the blue plane (gray and orange arrows).

**Figure 3 jpm-13-00549-f003:**
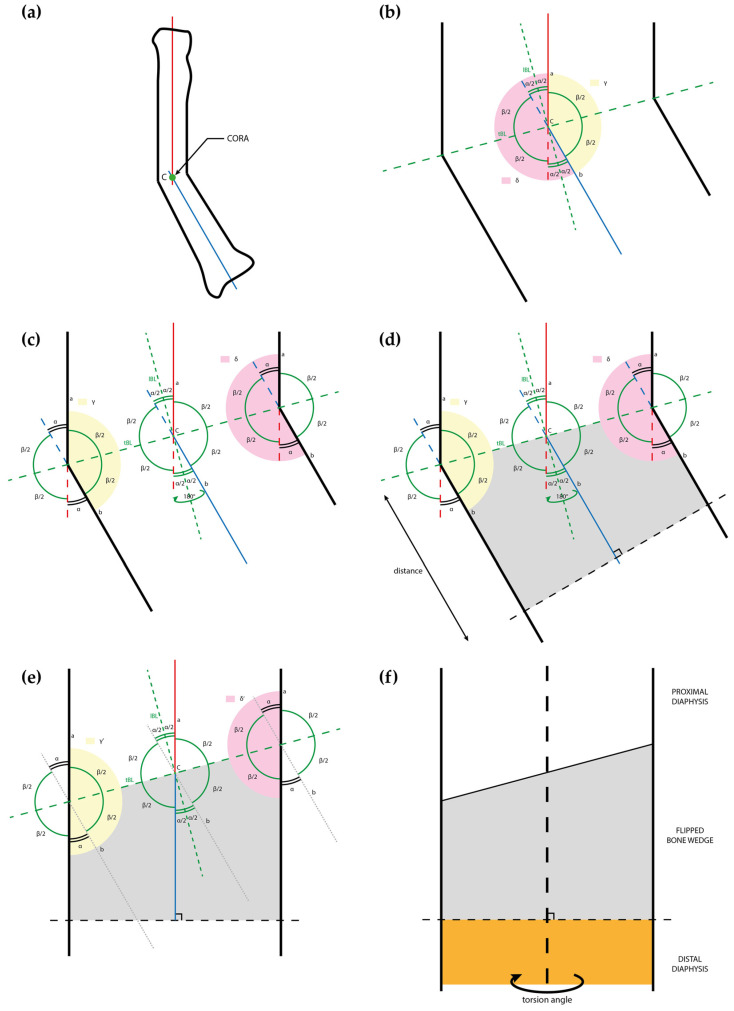
Drawings representing the steps of the flipping-wedge osteotomy and its geometrical demonstration: (**a**) drawing of a deformed radius sectioned along the plane of maximal deformity with the proximal and distal axes (respectively red and blue), which meet at the center of rotation of angulation (CORA); (**b**) at the CORA, the angular deformity (α) and its supplementary angle (β) are identified, and the respective longitudinal bisector line (lBL) and transverse bisector line (tBL) are drawn as dotted lines. A concavity angle (γ) and a convexity angle (δ) are colored in yellow and pink, respectively; (**c**) γ and δ overlap with the concave and convex shapes of the bone; (**d**) in this example, a bone wedge is identified (gray) between the tBL and a line perpendicular to *b* (black dotted line) at any desired distance from the CORA; (**e**) the gray wedge is flipped 180° and, according to the Proof of Theorem 1, the new concavity (γ’) and convexity (δ’) angles are both equal to 180°; and (**f**) rotational alignment can be further adjusted on the perpendicular plane, in this example by freely rotating the orange segment distal to the perpendicular cut. CORA = center of rotation of angulation; lBL = longitudinal bisector line; tBL = transverse bisector line.

**Figure 4 jpm-13-00549-f004:**
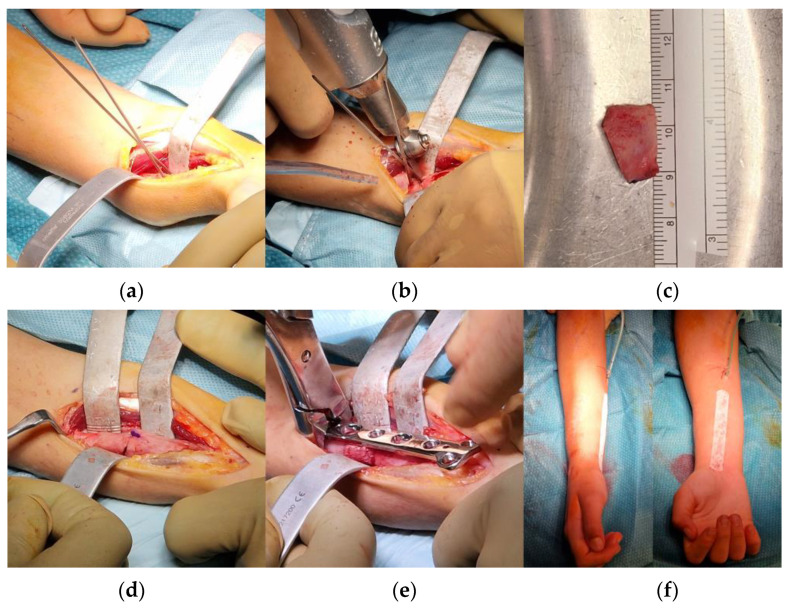
Steps of the surgical procedure: (**a**) according to preoperative planning, through a volar approach, two K-wires were positioned along the plane of maximal deformity from the concave side of the bone deformity; (**b**) guided by K-wires, two cuts perpendicular to the plane of maximal deformity were performed; (**c**) the desired bone wedge was obtained; (**d**) the bone wedge was flipped 180° on the longitudinal axis of the radius and used as an intercalary autograft for the osteotomy; (**e**) a locking-compression T-plate was implanted for fixation; and (**f**) the alignment of the forearm appeared acceptable at the end of the procedure.

**Figure 5 jpm-13-00549-f005:**
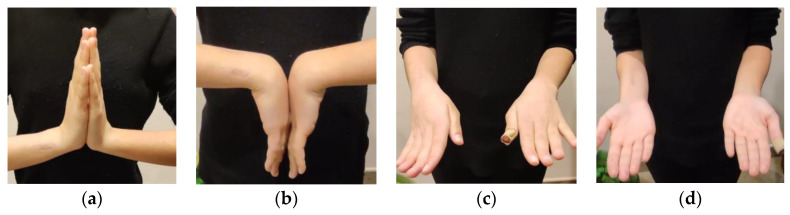
Postoperative pictures at 5-month follow-up documenting (**a**,**b**) wrist and (**c**,**d**) forearm range of motion.

**Figure 6 jpm-13-00549-f006:**
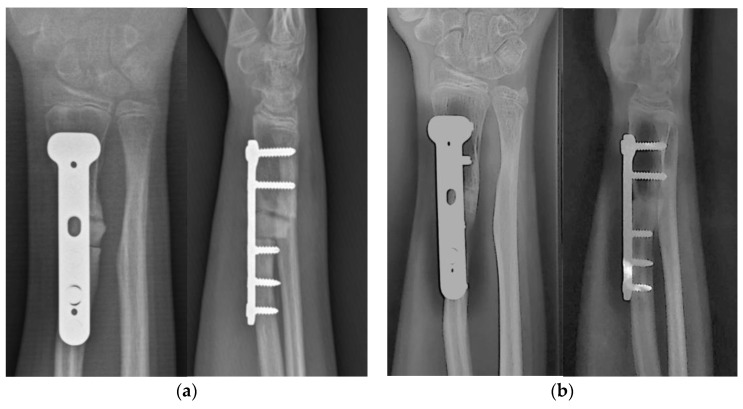
Postoperative radiographic images (**a**) at 3-month follow-up and (**b**) at 5-month follow-up, showing progression of consolidation at the osteotomy site and neutral ulnar variance.

## Data Availability

Data are available from the corresponding author upon reasonable request.

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
