# Peer review of "The Flipping-Wedge Osteotomy: How 3D Virtual Surgical Planning (VSP) Suggested a Simple and Promising Type of Osteotomy in Pediatric Post-Traumatic Forearm Deformity"

_jpm, 2023, doi:10.3390/jpm13030549_

Round 1

Reviewer 1 Report

The presentet article is very interesting. The topic and the presentation is outstandig.

Author Response

Dear Reviewer,

Thank you very much for your time and your appreciations.

Kind Regards.

Alessandro Depaoli

Reviewer 2 Report

1. authors write: " Initially, some surgical plannings were simulated with the aim to empirically align the malunited radius to the uninjured one by reallocating a local bone wedge, as it was done in the other case" - what is the difference between this paper and the previous one? It should be clearly stated, even underlined !!!!

2. Conclusions are not supported by results (the paper is about one case, and in conclusions you mentioned two patients?)

Author Response

Dear Reviewer,

1. Thank you for the suggestion. We already showed in a previous article a case of post-traumatic deformity of the proximal radius that was treated by repositioning a local bone wedge. In that case, the correction was planned on the 3D model, using a time-consuming “trial-and-error” process and no predetermined geometric rules were established. In the present case, we applied and demonstrated a geometrical approach that allowed us to plan an accurate correction with a reproducible and time-saving method. However, to avoid any ambiguity, we decided to not discuss that previous case.

2. The manuscript was unclear and we changed the conclusion only with the case described.

Thank you for your time and your precious suggestions.

Kind Regards,

Alessandro Depaoli, MD

Reviewer 3 Report

First of all, I would like to congratulate you for successfully performing surgery on a difficult case.

1) How many cases have you had surgery with this technique?

2) Can this technology work well for adults? Although this case was a child, there is a clear possibility of non-union when applied to adults. Therefore, readers should be informed of this.

3) After step C in Figure 2, was there any case where a bone graft was needed due to an additional ostectomy or defect?

4) Is this patient's last follow-up 5 months? When was complete bone union achieved?

5) Please present a photo of the surgical site and range of motion photo at the final follow-up.

6) Is there a functional analysis of the patient? The fact that there is no functional analysis after surgery also seems to be a problem.

Author Response

Dear Reviewer,

1.Up to now, just one patient with forearm deformity has been fully treated with the exact geometrical rules of this technique, from the planning to the final procedure. We cited in the text an article in which some authors of this manuscript already reported a case of correction of post-traumatic forearm deformity treated by repositioning a bone wedge, which was planned in a totally empiric way on the 3D model by engineers with no defined geometric rules. However, we decided to remove this case to avoid any possible ambiguity in the text.

2.We thank for the suggestion and, as already exposed in Discussion section, we agree that there is a high risk of non-union. We highlighted this point in the Discussion section in the text.

3.No. The technique proposed in the text aimed to fully correct deformity with the rotation of the most geometrically advantageous local bone wedge in order to preserve bone length.

4.The last follow-up was 5 months and we observed complete bone healing with good range of motion. Further radiographs were not requested.

5.We added the postoperative photos as suggested.

6.We added a detailed assessment of postoperative range of motions.

Thank you very much for your time and your suggestions.

Kind Regards.

Alessandro Depaoli, MD